# Quantification of Elemental Contaminants in Unregulated Water across Western Navajo Nation

**DOI:** 10.3390/ijerph16152727

**Published:** 2019-07-31

**Authors:** Jonathan Credo, Jaclyn Torkelson, Tommy Rock, Jani C. Ingram

**Affiliations:** 1College of Medicine Clinical Translational Science Graduate Program, University of Arizona, Tucson, AZ 85721, USA; 2Department of Chemistry & Biochemistry, Northern Arizona University, P.O. Box 5698, Flagstaff, AZ 86011, USA

**Keywords:** unregulated water, Navajo, arsenic, uranium, manganese

## Abstract

The geologic profile of the western United States lends itself to naturally elevated levels of arsenic and uranium in groundwater and can be exacerbated by mining enterprises. The Navajo Nation, located in the American Southwest, is the largest contiguous Native American Nation and has over a 100-year legacy of hard rock mining. This study has two objectives, quantify the arsenic and uranium concentrations in water systems in the Arizona and Utah side of the Navajo Nation compared to the New Mexico side and to determine if there are other elements of concern. Between 2014 and 2017, 294 water samples were collected across the Arizona and Utah side of the Navajo Nation and analyzed for 21 elements. Of these, 14 elements had at least one instance of a concentration greater than a national regulatory limit, and six of these (V, Ca, As, Mn, Li, and U) had the highest incidence of exceedances and were of concern to various communities on the Navajo Nation. Our findings are similar to other studies conducted in Arizona and on the Navajo Nation and demonstrate that other elements may be a concern for public health beyond arsenic and uranium.

## 1. Introduction

The geology of the Four Corners region of the American Southwest is comprised of a sandstone and limestone bedrock with an iron-oxide and iron-sulfide mineralite matrix that contains an abundance of natural resources, including coal, copper, uranium, and vanadium [1,2,3]. For this reason, mining in the Four Corners region has been ongoing for the past 100 years and has left a legacy of widespread environmental contamination, affecting groundwater and soil near mine features. In addition to contamination from mining and ore refining, the iron-oxide and iron-sulfide mineralite matrix present an added hazard due to their association with heavy arsenic concentrations [4,5]. Natural arsenic is bound loosely through ionic interactions with a variety of mineralites, including iron-oxides and iron-sulfides, which can be released into the surrounding environment. Mining practices expedite this process by bringing ores closer to the surface and increasing the surface area of these iron-arsenic species, as well as concentrating waste through tailing piles and run-off [6,7].

Rural and tribal communities are particularly vulnerable to environmental contamination due to a lack of public infrastructure, insufficient medical facilities, and low socioeconomic status [8,9]. Additionally, a higher density of large-scale mining and refining operations are found in these communities, which increases the risk of exposure and contamination [10,11,12]. One example is the Navajo Nation, located within the Four Corners region of the American Southwest. From 1944 through 1986, the Navajo Nation was the largest producer of domestic uranium ore in the United States, primarily for use in nuclear munitions during the Cold War [13,14,15,16]. Uranium mining exposures have been directly linked to an increase in lung and other uranium related cancers in miners and their families [14,17,18]. Though uranium mining ceased on the Navajo Nation in 1986, the contamination left from these operations still represents a significant danger to the Navajo people [16,19]. Unregulated “Livestock Only” water sources found across the Navajo Nation are susceptible to contamination from previous mining operations, and their unregulated nature can result in an exceedance of various United States Environmental Protection Agency (EPA) Maximum Contamination Levels (MCL) [20,21,22,23]. These unregulated water sources often represent the most convenient source of water for human consumption, household use, and watering crops for families and communities across the Navajo Nation. In many cases, the closest regulated water source for water hauling is at least an hour’s drive [12].

Both arsenic and uranium’s health effects from ingestion of contaminated drinking water have been widely established. Arsenic is recognized as a known carcinogen and has been demonstrated to cause vascular damage like that seen in chronic heart disease. Additionally, studies have demonstrated that arsenic can play a significant role in the development of chronic diseases by acting as a potentiating agent in concert with other contaminants [24,25,26,27]. Uranium as a contaminant in drinking water has been demonstrated to accumulate in the kidneys causing progressive kidney damage, which can lead to renal compromise or compound kidney damage related to diabetes. Its link to cancer, especially with Navajo uranium miners, was primarily tied to inhalation of uranium dust and radioactivity [7,28,29].

Numerous papers have reported that both arsenic and uranium frequently exceed the EPA’s MCLs in unregulated water sources on the Navajo Nation [30,31,32,33]. These studies further demonstrated that arsenic contamination is more widespread than that of uranium contamination. In one study by Hund and colleagues, they showed that there is a strong association between arsenic and uranium and suggested the existence of a belt of contamination running north to south down the central portion the Navajo Nation [34]. Although a plethora of studies and information exist on arsenic and uranium contamination on the New Mexico side of the Navajo Nation, the same scale of information does not exist on the Arizona or Utah side. In these regions, studies have focused on specific areas of the Navajo Nation, while leaving the surrounding areas as an unknown. Furthermore, water quality studies on Navajo thus far have focused primarily on arsenic and uranium contamination. There has been recent interest in ascertaining the possible contamination of unregulated water systems on the Arizona and Utah side of the Navajo Nation from other common contaminants, including copper, lead, manganese, and mercury, as part of studies to determine the risks of possible future uranium mining around the Grand Canyon [35,36]. This paper seeks to address two questions- the first is to determine if arsenic and uranium contamination is as widespread on the Arizona and Utah side of the Navajo Nation as demonstrated on the New Mexico side, and second to quantify the extent of other possible elemental contamination that may exist. The information presented in this paper will serve to provide information to Navajo Nation leaders, as well as affected communities, on the quality of contamination of their water systems and will serve as baseline information for elements that, up to this point, have not been fully explored.

## 2. Materials and Methods

### 2.1. Study Area

The Navajo Nation is the largest contiguous Native American reservation in the continental United States. Located within the Four Corners region of the American Southwest, its borders span 71,000 square kilometers across Arizona, New Mexico, and Utah. The Navajo Nation is recognized by the United States’ government as a sovereign nation, though the United States retains plenary power and is separated into 110 tribal Chapters governed through five management Agencies: Chinle (14 Chapters), Crownpoint/Eastern (31 Chapters), Ft. Defiance (27 Chapters), Shiprock (20 Chapters), Tuba City/Western (18 Chapters). The Navajo Nation is within the Colorado Plateau region where the climate is largely controlled by orographic effects and elevation. Areas below 1370 m (4500 ft) are semiarid. The average precipitation is 20 to 30 cm on average per year. However, some lowland areas may receive less than eight centimeters of precipitation per year. Most of the Navajo Nation is in a rain shadow where much of the precipitation comes from the south and is blocked by the southern rim of the Colorado Plateau. Up to 65% of the yearly precipitation occurs during the late summer months (July and August) and can result in flash flooding. All runoff goes to the Colorado River, either directly or via one of the tributaries (the San Juan and the Little Colorado Rivers) [37].

In the western portion of the Navajo Nation, rocks from the Cretaceous Dakota formation and below are present. However, regional erosion patterns have resulted in progressively older rocks being exposed at the surface in the southwest portion of the Navajo Nation [38]. Recharge of the aquifers occurs in upland areas, which divide the land into separate hydrologic basins. There are five distinct hydrologic basins, which are Black Mesa, San Juan, Blanding, Henry, and Kaiparowits. Water that is recharged in the upland areas moves downward towards the major rivers and tributaries [37].

The main sources of groundwater for the Navajo Nation come from the Navajo (N) aquifer, the Coconino (C) aquifer, and shallow alluvium aquifers [37]. The N aquifer is an important groundwater source in areas north of the Little Colorado River, and water quality is considered relatively good except in areas where past uranium mining and milling occurred [39]. Formations of the N aquifer include the Jurassic Navajo Sandstone, Kayenta Formation, and Lukachukai Member of the Wingate Sandstone. These formations are hydraulically connected and act as a single aquifer [40]. The N aquifer receives recharge in areas near Shonto where Navajo Sandstone is exposed at the surface. In other parts of Black Mesa, the N aquifer has overlying confining layers, which limit recharge [41]. Groundwater that is recharged near Shonto flows radially in the southwest direction to Tuba City, as well as to the south and east [40].

The C aquifer is an important groundwater source south of the Little Colorado River. North of the river, the C aquifer is too deep to access, and the high level of salinity (total dissolved solids) make it undesirable to use for a drinking water source [39]. The C aquifer includes the Pennsylvanian and Permian Upper and Middle Supai Formations, the Permian Coconino Sandstone, and the Permian Kaibab and Schnebly Hill Formations [42].

The map in Figure 1 displays the borders of the Navajo Nation and outlines the study area. Samples included in this study are represented in yellow and previous sampling efforts by the Centers for Disease Control and Prevention (CDC), the United States’ Environmental Protection Agency (EPA), and Army Corps of Engineers are also provided [43]. Abandoned uranium mines are displayed in this map, represented by circles of differing size, color, and score based on EPA’s Hazard Ranking System used for assessing a contaminated site’s position on the National Priorities List. Locations with a higher score (i.e., 17,640) are larger and darker and deemed to pose a greater threat to human health [44]. The present study area was restricted to the Arizona and Utah Western, Chinle, Shiprock, and Fort Defiance Navajo Management Agencies.

### 2.2. Sample Collection

Water samples (*n* = 296) were collected from a variety of unregulated groundwater sources accessed by windmills, troughs, springs, and water storage tanks. Samples were identified from working with the Navajo Tribal Utility Authority (NTUA) branch offices, previous surveys conducted by the U.S. Army Corps of Engineers (US EPA 2000a), along with community and chapter members. The unregulated nature of these water sources means that they are not regularly monitored for bacterial or elemental contamination and are not upheld to the same Clean Drinking Water Act standards, despite often being an important, if not the only, source of water for communities. Their unregulated designation is due to these water sources falling outside of set active management areas as defined by the Arizona Department of Water Resources and the Navajo Nation Department of Water Resources [45,46].

Sampling was conducted during January–March and June–August of 2014, January–March of 2015, July 2016, and June and October 2017, with many sites being sampled numerous times for the evaluation of seasonal fluctuation and verification of elemental concentrations (see Appendix A). Upon arrival to each site, the GPS coordinates were recorded using WGS84 Map Datum, a site description, including the presence of flora and fauna and notable features, type of water source, and site pictures were collected. An identification code was assigned to each site based on previously labeled designations, if available, the name was given by community members or readily identifiable features. Water sources were turned on and ran for one to two minutes before collection to ensure accurate representation. Water sources were not run for two minutes before collection in the presence of community members to avoid concerns regarding water waste or if water source levels were low upon visual inspection of the storage tank to avoid depleting the resource for community use. Samples for elemental analysis were filtered through a 0.45-micron filter, preserved in nitric acid (HNO_3_—Omnipure grade) to a pH below 2, and sent back to Northern Arizona University (NAU) to be stored at 4 °C until analysis [47]. Samples for total mercury analysis were acid preserved in hydrochloric acid (HCl) to a pH below 2 and sent back to NAU to be stored at 4 °C until digestion and analysis [48].

### 2.3. Sample Analysis and Instrumentation

The analysis was conducted by inductively coupled plasma mass spectrometry (ICP-MS) (Table 1), inductively coupled plasma optical emission spectroscopy (ICP-OES) (Table 2), flame atomic absorption (FAA), or cold vapor mercury atomic absorption (CVAA) for a variety of elements. All samples analyzed by ICP-MS and ICP-OES were diluted 1:10 with a dilution solution of 2% HNO_3_ that contained an internal standard corresponding to the instrument and analysis method used. A Thermo Fisher Scientific X-Series 2 ICP-MS (Northern Arizona University, Flagstaff, AZ, USA) was used for all analyses with an ESI APEX HF dissolving nebulizer introduction system (Northern Arizona University, Flagstaff, AZ, USA) to produce higher sensitivity. Before each analysis, an auto-tune sequence and stage alignment were conducted followed by a manual tune for As or U signals to optimize detection and maintain stability throughout the analysis. During element selection, both the most common and most stable masses were selected, and the resulting concentrations compared; lead had numerous masses selected to accurately quantitate its occurrence in the sample given the differing radiogenic origins [49,50]. To overcome the isobaric interference from the ^75^ArCl polyatomic species and produce accurate arsenic detection, an internal correction equation was programed on the ICP-MS utilizing ^77^ArCl, ^82^Se, ^83^Kr signals [51]. To further limit isobaric interferences of polyatomic ions and optimize detection, analysis on the ICP-MS was separated into “light” and “heavy” elements for analysis. ICP-OES analysis was conducted with a Perkin-Elmer Optical Emission Spectrometer Optima 4300 DV (Northern Arizona University, Flagstaff, AZ, USA) with standard plasma settings and 2-point background correction. To overcome spectral interferences from overlapping emission lines, the analysis was split into two separate methods.

A Perkin Elmer Analyst 200 Flame Atomic Absorption Spectrometer (Northern Arizona University, Flagstaff, AZ, USA) with standard operating settings was used to determine total calcium in collected samples. Samples were diluted either 1:10 or 1:100 with a 2% HNO_3_ solution. Also, an aliquot of both lanthanum nitrate and cesium chloride were added as matrix modifiers and ionic suppressants to a final concentration of 1000 mg/L. Similar quality assurance and quality control methods from ICP-MS and ICP-OES analysis were applied to FAA analysis [52].

To evaluate total mercury in the samples, a Perkin-Elmer FIMS 100 Cold Vapor Mercury Analyzer (Northern Arizona University, Flagstaff, AZ, USA) with standard operating settings and procedures was used. Field samples were digested with an aliquot of 1 mL 5% (w/v) KMnO_4_ in 0.1% HCl and placed in an 80 °C water bath for an hour. Before analysis, hydroxylamine hydrochloride and an antifoaming agent were added to quench excess KMnO_4_. To ensure quality assurance and quality control, a lab standard and certified reference material were run alongside the samples [53].

### 2.4. Quality Assurance/Quality Control

To ensure accuracy of the analysis, a National Institute of Standards and Technology Standard Reference Material 1640a (NIST 1640a), field and instrument blanks, and check standards were analyzed alongside the samples and were periodically re-analyzed throughout all analyses. Additionally, a split subset of 121 samples were sent to two separate laboratories, the Arizona Laboratory for Emerging Contaminants at the University of Arizona in Tucson, Arizona and the University of New Mexico’s Analytical Chemistry Laboratory in Albuquerque, New Mexico, to have 20 of the investigated elements analyzed to compare the analysis conducted at NAU.

## 3. Results and Discussion

Water samples from unregulated sources on the Navajo Nation were sampled from 2013 through 2017 and were analyzed for 21 elements by ICP-MS, ICP-OES, FAA, and CVAA. The concentrations of each element were compared to US EPA MCLs [20,54] and US national averages reported by Agency for Toxic Substances and Disease Registry (ATSDR) [55], United States Geological Society (USGS) [56], the World Health Organization [57], and Morr [58]. Although these national regulatory guidelines are for regulated and treated drinking-water supplies, these limits serve as an appropriate basis for comparison for unregulated and untreated water used for human consumption. Table 3 displays the total number of samples for each element that had a reported value that exceeded, approached, was below the guidelines, or below detection (BD) by the method used (see Appendix A). A sample was “approaching” a limit if its concentration was within 75% of the reported regulatory limit or national average, which has been suggested as a precautionary guideline to monitor water resources [59,60,61,62,63,64].

Of the elements analyzed, only 14 elements had samples that had at least one instance of a concentration greater than either a national regulatory limit or national averages (Table 3). Of these 14 elements, six (V, Ca, As, Mn, Li, and U) were selected to expand upon either because of many samples had concentrations greater than a guideline or based on the risk of detrimental human health impacts [55].

For arsenic analysis, 296 sites were visited, and 61 sites were dry; and for uranium analysis, 297 sites were visited, and 66 sites were dry. Of the remaining sites, these were sampled at least twice, the results from the analysis combined, and the averages reported (Table 4). The other 19 elements were conducted on a subset of 121 or 124 samples, which were also surveyed at least twice, and the averages of the analysis reported (Table 4). On comparing the averages to the guidelines, we found that V, Ca, and Mn concentrations were above these limits, As and Li were approaching, and U was below. All the elements, except for Hg, had an outlier with a concentration value that exceeded the guideline (Table 4). The effect of evaporation, different water storage containers, seasonal variability, and other variables likely influence the concentration of the contaminants and warrants further investigation. 

Vanadium had the greatest number above comparative guidelines at 97 of 121 samples collected, whereas U had the lowest prevalence of samples above comparative guidelines at 21 of 231 samples collected (Table 3). To ascertain if these six elements were associated with specific regions on the Navajo Reservation, the results were separated by the four agencies from which the samples were collected: Chinle, Fort Defiance, Shiprock, and Western Agency (Table 5). Fort Defiance and Western Agency had the highest exceedances for As, U, and V. Chinle and Western Agency had the highest exceedances for Ca, Li, and Mn.

A combination of a long history of hard rock mining, limited water infrastructure, and a natural geologic profile high in metal and metalloid species increases the potential for drinking water contamination on the Navajo Nation. An added factor for concern is that as much as 30% of those living on the Navajo Nation get their drinking water from unregulated sources [68]. The results presented in this study demonstrated that there are comparable levels of arsenic in unregulated water resources to rural communities across the United States [5,69]. Arsenic was detected at concentrations greater than the US EPA MCL of 10 μg/L in 17% (40 out of 235) of sampled sites. Welch and colleagues found slightly less than half of 30,000 arsenic analyses of groundwater in the United States at 1 μg/L and about 10% exceeded 10 μg/L [5]. However, arsenic concentrations greater than 10 μg/L were more frequently observed in the western United States than in the eastern half. Looking specifically at Arizona and the Navajo Nation, these results reveal comparable concentrations and incidences of regulatory infractions for elemental contaminants, especially arsenic [31,70]. Jones and colleagues utilized public water databases to quantify arsenic concentrations across Arizona and found that rural areas and those associated with mining ventures had arsenic in water systems at similar concentrations to those represented in this study, but it lacked information about the Navajo Nation [70]. Hoover and colleagues combined federal databases and sampling efforts by the University of New Mexico to characterize the extent of elemental contamination in drinking water across the Navajo Nation [31]. Like this study, they found elemental contamination existed across the Navajo Nation and was closely dependent on the agency where sampling was conducted. The variability of elemental contamination in water noted in this study and the Hoover study may be related to differing access to water across the Navajo Nation [31].

Beyond arsenic and uranium, the results demonstrate an elevation above US EPA guidelines and USGS average water content of four other metals (Ca, Li, Mn, V). These four metals have been investigated in only a few previous publications, the most significant being a spatial clustering analysis by Hoover (2018) that matched co-eluting contaminants in water sources across Navajo [52,71,72]. Elevations of calcium and vanadium in drinking water do not seem to have any significant detriment other than changes in the aesthetic quality of the water, and these elevations seem reasonable given the geologic profile of the Navajo Nation [56,65,73]. Some studies suggest lithium in drinking water is associated with decreased instances of some mental health conditions (depression, anxiety, and dementia); however, this is still hotly debated. Additionally, lithium exposure has been linked with the development of chronic kidney disease [72,74,75,76,77]. There is growing evidence that chronic exposure to manganese in drinking water may have similar neurotoxic effects as occupational exposure to manganese, resulting in intellectual impairment in children and visible brain deposition in mouse models [78,79,80]. Bouchard and colleagues in 2007 conducted a pilot study investigating manganese exposure through tap water and the incidence of hyperactivity as a pathway for learning impairment in Canadian children [81]. Their results demonstrated at concentrations, like what was seen in the present study, were correlated with more cases of hyperactive behavior and outbursts during classroom settings. They followed up this study in 2011 with a cross-sectional study examining IQ scores in children and manganese exposure through water [82]. Much like their earlier study, Bouchard and colleagues found that elevated levels of manganese in drinking water seemed to correlate with decreased IQ scores, and these findings have been repeated in other studies from other groups [83,84,85]. While the concentrations of manganese in the Canadian schoolchildren study conducted by Bouchard were slightly higher, an average of 200 μg/L, Bouchard asserts that the children exposed to these concentrations were not relying on these water sources as their sole supply, and it is more important to look at a cumulative dose. Compared to this study, while the median manganese concentration is lower, the lack of available water in the areas sampled in this study may have a similar cumulative effect mentioned by Bouchard and is an avenue of study that warrants further investigation. Compared to arsenic and uranium in drinking water, both of which are recognized primary drinking water contaminants, elevated levels of calcium, lithium, manganese, and vanadium are based on non-enforceable guidelines or national averages and, in the case of manganese, these limits have been abolished. Despite this, these guidelines, as with primary standards, are a metric for which to compare the drinking water concentrations. Furthermore, arguments question if the current guidelines used to create enforceable regulatory limits on drinking water are insufficient, especially given research that opposes the previous dogma and supports observable health consequences to exposed populations [59,86].

## 4. Conclusions

This paper sought to address two questions- the first was to determine if arsenic and uranium contamination is as widespread on the Arizona and Utah side of the Navajo Nation as demonstrated on the New Mexico side, and second to quantify the extent of other possible elemental contamination that may exist. The results reported here have demonstrated similarities in drinking water contaminants across the Navajo Nation, the American Southwest, and the United States; they have additionally demonstrated the potential for other contaminants that may pose measurable health effects in those exposed. A hydrogeochemical approach may expound upon the findings in this study, as well as others, especially elucidating the reason why certain regions of the Navajo Nation have a similar water contamination profile despite differences in mining history and available infrastructure. Clean up of abandoned uranium mines by the US EPA and other US government entities is moving into the second Five-Year Plan [87]. This timely study contributes to the current understanding of water quality on the Navajo Nation and will help guide future policy and clean up decisions, as well as contribute to the understanding of the potential for health impacts from exposure to these contaminants.

## Figures and Tables

**Figure 1 ijerph-16-02727-f001:**
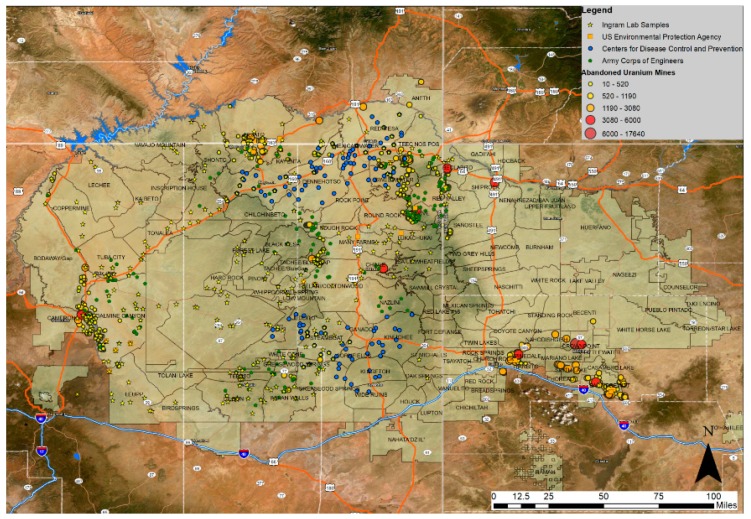
Map of the Navajo Nation. Sites of abandoned uranium mines are represented by circles, with larger and darker colored circles noting areas posing a greater threat to human health. Samples collected for this study (*n* = 296) are represented in yellow, denoted by the label “Ingram Lab Samples”, and previous sampling efforts of government agencies are displayed in green (Army Corps of Engineers), blue (CDC), and orange (U.S. EPA). CDC: Centers for Disease Control and Prevention; EPA: Environmental Protection Agency.

**Table 1 ijerph-16-02727-t001:** Summary of ICP-MS (inductively coupled plasma mass spectrometry) Analysis.

ICP-MS Analysis			
Element	Mass	Internal Standard	Light or Heavy
Cr	52	Rh	Light
Ni	58, 60	Rh	Light
As	75	Ir	Heavy
Mo	95, 98	Rh	Light
Cd	111, 114	Ir	Heavy
Sn	118, 120	Ir	Heavy
Sb	121	Rh	Light
Pb	206, 207, 208	Ir	Heavy
U	238	Ir	Heavy

**Table 2 ijerph-16-02727-t002:** Summary of ICP-OES (inductively coupled plasma optical emission spectroscopy) Analysis.

ICP-OES Analysis			
Element	Wavelength (nm)	Axial or Radial	Internal Standard
Ba	455.403	Radial	Y
Be	313.042	Radial	Y
Cu	325.747	Radial	Y
Fe	259.933	Radial	Y
Mn	257.604	Radial	Y
V	292.399	Radial	Y
Zn	213.855	Axial	Y

**Table 3 ijerph-16-02727-t003:** List of 21 elements analyzed by ICP-MS (inductively coupled plasma mass spectrometry) and several samples categorized concerning guidelines. Guideline sources: US EPA (Environmental Protection Agency) Unregulated Contaminant Monitoring Rule (UCMR) [65]; United States Geological Society (USGS) [56]; Morr (2006) [58]; US EPA Maximum Contaminant Level (MCL) [20]; US EPA Regional Screening Levels (RSL) [66]; US EPA National Secondary Drinking Water Regulations (NSDWR) [54]; US EPA Drinking Water Standards and Health Advisories (DWSHA) [67]; Agency for Toxic Substances and Disease Registry (ATSDR) [55]; World Health Organization (WHO) [57]; US EPA Maximum Contaminant Level Goal (MCLG) [20]; US EPA Action Limit (AL) [20]. * Lower limit was used for comparison.

Element	Guideline	Source of Guideline	Above	Approaching	Below	Below Detect	Total Water Sources Sampled
V	21.0 μg/L	US EPA UCMR	97	2	18	4	121
Ca	21.8 mg/L	USGS and Morr	69	4	51	0	124
As	10 μg/L	US EPA MCL	40	6	157	32	235
Mn	50 μg/L	US EPA NSDWR	29	1	91	0	121
Li	40 μg/L	US EPA RSL	56	15	50	0	121
U	30 μg/L	US EPA MCL	21	10	179	21	231
Al	50–200 μg/L *	US EPA NSDWR	13	3	105	0	121
Mo	80 μg/L	US EPA DWSHA	7	3	103	11	124
Sr	4 mg/L	US EPA DWSHA	7	4	110	0	121
Fe	300 μg/L	US EPA NSDWR	4	6	110	1	121
Ni	3-10 μg/L *	ATSDR	3	16	100	5	124
Sn	1.1–2.2 μg/L *	WHO	1	5	19	99	124
Be	4 μg/L	US EPA MCL	1	0	42	78	121
Cd	5 μg/L	US EPA MCL	1	0	9	114	124
Zn	5 mg/L	US EPA NSDWR	0	1	120	0	121
Hg	2 μg/L	US EPA MCL	0	0	124	0	124
Cu	1.3 mg/L	US EPA MCLG	0	0	121	0	121
Ba	2 mg/L	US EPA MCL	0	0	121	0	121
Cr	100 μg/L	US EPA MCL	0	0	105	19	124
Sb	6 μg/L	US EPA MCL	0	0	106	18	124
Pb	15 μg/L	US EPA AL	0	0	78	46	124

**Table 4 ijerph-16-02727-t004:** List of 21 elements analyzed by ICP-MS (inductively coupled plasma mass spectrometry) reporting the maximum, minimum, average, and median values from the set of samples listed in Table 1. B.D. = Below Detect.

Element	Max (μg/L)	Min (μg/L)	Average (μg/L)	Median (μg/L)
V	520	B.D.	81.70	67.30
Ca	430	0.35	44.7	25.6
As	190	0.03	8.21	1.99
Mn	14700	0.10	164	3.44
Li	630	3.02	63.3	37.6
U	490	0.04	14.1	3.05
Al	64600	2.16	556	12.1
Mo	1190	B.D.	27.2	2.89
Sr	10300	18.9	1160	478
Fe	605	0.02	61.5	23.1
Ni	560	0.02	6.42	1.09
Sn	2.50	0.01	0.53	0.40
Be	60.3	B.D.	0.50	0.00
Cd	11.1	0.01	0.30	0.05
Zn	3900	3.38	197	48.1
Hg	B.D.	B.D.	B.D.	B.D.
Cu	26.0	0.02	2.80	1.21
Ba	1200	7.91	177	93.7
Cr	12.1	0.03	0.94	0.47
Sb	2.80	0.03	0.33	0.27
Pb	9.25	0.02	0.66	0.10

**Table 5 ijerph-16-02727-t005:** Distribution of As, U, V, Mn, Ca, and Li concerning guideline comparisons across the Chinle, Fort Defiance, Shiprock, and Western Agencies. B.D. = Below Detect.

		Agency				
Element	Guideline	Chinle	Fort Defiance	Shiprock	Western	Totals
Arsenic	Above	3	23	3	11	40
	Approaching	0	4	0	2	6
	Below	31	29	10	89	159
	B.D.	11	2	2	15	30
	Totals	45	58	15	117	235
Uranium	Above	3	7	1	7	18
	Approaching	3	3	0	5	11
	Below	32	44	14	93	183
	B.D.	6	2	0	11	19
	Totals	44	56	15	116	231
Vanadium	Above	24	26	7	40	97
	Approaching	0	0	0	2	2
	Below	5	1	1	11	18
	B.D.	2	2	0	0	4
	Totals	31	29	8	53	121
Manganese	Above	19	4	1	5	29
	Approaching	1	0	0	0	1
	Below	12	25	7	47	91
	B.D.	0	0	0	0	0
	Totals	32	29	8	52	121
Calcium	Above	26	11	1	31	69
	Approaching	0	0	0	4	4
	Below	8	18	6	19	51
	B.D.	0	0	0	0	0
	Totals	34	29	7	54	124
Lithium	Above	19	13	5	19	56
	Approaching	6	5	1	3	15
	Below	6	12	2	30	50
	B.D.	0	0	0	0	0
	Totals	31	30	8	52	121

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
