# Peer review of "Quantification of Elemental Contaminants in Unregulated Water across Western Navajo Nation"

_ijerph, 2019, doi:10.3390/ijerph16152727_

Round 1

Reviewer 1 Report

The methods used for sampling and analysis were sound, but the interpretation of the results relative to benchmarks (referred to as "regulatory limits") had several flaws that warrant revision and re-review of this manuscript. 

Lines 17-18 and elsewhere (line 48, etc.): The expression "exceeding a national regulatory limit" warrants clarification in the main body of the report for two reasons; (1) if these waters are unregulated, how are regulatory limits applicable? how are these limits to be used? (2) the term "exceeding" in this context is misleading, since in the legislative context, these regulatory limits are defined as applying to treated drinking water that is delivered to customers; therefore, any concentrations in environmental waters that are "greater than" these concentration limits are not truly "exceedances" in the regulatory context -- that is, the regulations have not been violated or exceeded because they do not apply to unregulated or untreated waters. Consider providing an explanation, or else try rewording as "concentrations greater than the regulatory limit" to avoid asserting that exceedances have occurred. If families in the Navajo Nation are consuming untreated water from these sources, that's an important point to make. I think you can explain that although the EPA MCLs are intended for ensuring the safety of regulated drinking-water supplies, which typically undergo treatment prior to delivery to customers, these limits are an appropriate basis for comparison for the unregulated waters sampled for your study, as unregulated and untreated water sources are used for human consumption.

Line 44: "Though uranium mining ceased in the 1980s..." warrants clarification -- is that end date accurate? Do we know precisely when in the 1980s? Was a particular law enacted that can be cited that firmly substantiates an end date to the mining? I found a reference that states, "From 1944 to 1986, an estimated 3,000 to 5,000 Navajo people worked in the uranium mines on their land.[11]" (Fettus, Geoffry H.; Matthew G. Mckinzie (March 2012). "Nuclear Fuel's Dirty Beginnings: Environmental Damage and Public Health Risks From Uranium Mining in the American West" (PDF)National Resources Defense Council. National Resources Defense Council. Retrieved 29 April 2014.). Also "In 2005, the President of the Navajo Nation, Joe Shirley, Jr., signed the Diné Natural Resources Protection Act which banned uranium mining and processing on Navajo land." Also, have any mining activities been restarted outside of Navajo lands that could potentially adversely affect the water resources in the Navajo Nation? 

Line 64-65: "a belt of contamination down the central portion the Navajo Nation..." -- I would like to know the direction or orientation of this belt of contamination, but I don't feel like having to look it up in the reference. Can you please add the missing information from Hund and others (2015). The word "down" implies a north-to-south orientation, but I would like to know for sure. (For example, "a belt of contamination running north to south down the central portion...").

Lines 72-73: "following the possible reopening of uranium mines around the Grand Canyon and the subsequent transport of uranium..." I am confused by this wording. Why is it "possible reopening"? Have mines been reopened or not? The phrase "subsequent transport" sounds like the contamination has already happened. It's unclear to me whether the "recent interest" (line 70) is in obtaining baseline data before the mines reopen or whether the mines have indeed already opened, but we're not sure if contamination has occurred. I can't tell which events are anticipated in the future and which events might have already happened.

Line 77: "information to regulatory agencies" -- to what end? The title of the article asserts that these waters are unregulated. What actions can the regulatory agencies take in response to information on unregulated waters? Likewise, what can the communities do? Will the information be used by the Navajo Nation to prioritize infrastructure improvements? As worded, it's not clear that the information from this study will be used for finding solutions, such as alternate supplies or clean-up technologies. I realize that this comment is a social concern (not technical), but I feel like this article contributes to an acceptance of the problem if it doesn't identify which parties are responsible for ensuring the availability of clean drinking water.

Line 98: What do the numbers next to the dots on the legend mean? Are they concentrations? (If so, what units?) Or are they numbers of mines? Please clarify. The yellow dots on the legend are labelled as "Ingram," but that's not explained anywhere... looks like an author's name, not a lab.

Line 105: Windmills are not a water body. Is there a better term to use? How does water get into (or onto?) a windmill? What is the source of the water? It is rainwater? Are the windmills connected to pumps deliver groundwater from local wells? Please clarify this. Perhaps it's obvious to people who live there, but it's not clear to readers who aren't local residents. Likewise, with the troughs and tanks, it would be helpful to know whether these sources are groundwater vs. rainwater or surface water, or a combination of sources. -- Tables 3 and 4 refer to numbers of "wells," and wells aren't even listed in this sentence.

Lines 110-113: If "livestock tanks" (or unregulated troughs) are the only source of drinking water for a community, is there an overlying social problem with these communities lacking appropriate sources of drinking water? See my previous comment regarding line 77. It seems wrong that a tank would be labeled "Livestock Only" if it's widely known that people drink that water because there is no alternative nearby. Even without the results of this study, we know that humans need clean drinking water that isn't labeled "livestock only." Am I understanding the situation correctly? Or is the human consumption of water from livestock tanks more of a matter of convenience (that is, because it's easier to drink water from a nearby tank than it is to carry clean water from elsewhere)? My concerns are nuanced, but what I am trying to say is that this article implies that there is a tacit acceptance of humans drinking water that is not appropriate for human consumption. Quantifying the contamination in the water is a very important finding because otherwise, those responsible for solving the problem (and perhaps the people who are drinking the contaminated water) are not taking the problem very seriously. 

Line 122: "delay in sample collection was not conducted if community members were present" -- why? Did the authors assume that if community members were present, it's likely that the water had been running already, and therefore was already representative? Please clarify. Otherwise, it sounds like there's no good reason to modify the sample-collection protocols merely because other people are nearby.

Line 123: Delete the word "total" from "Samples for total elemental analysis were filtered..." because "total" is widely understood to mean both the dissolved phase and the particulate phase (together or summed), and we don't get a "total" analysis from a filtered sample.

Lines 126-127: What was the reason for using two different treatments for the mercury samples?

Lines 158-161: Back in lines 126-127, I was only mildly curious, but now that it's come up again, I would really like to know -- What was the reason for using two different treatments for the mercury samples?

Line 165-166: Please clarify. Was only 1 analysis of the NIST 1640a standard performed throughout the study? Or can something more informative be said about the numbers of each type of QC sample relative to the numbers of environmental samples?

Line 167: Do you mean "split" samples? Or were these sequential replicates? ("Additionally, a subset of 121 samples") -- How were these samples divided or split, or were they replicates?

Line 168: "Certified" by whom? (Identify which organization did the certifying, please.)

Lines 175-177: What are the national averages reported by ATSDR (2005) from? Likewise, the USGS data (2016) and Morr (2006)? Are these averages in environmental water samples or some other matrix? These three sources appear to have no reference numbers, so it was difficult for me to figure out what they were. 

ATSDR: The ATSDR reports averages for crustal concentrations, air, and food consumption, not just water, so readers need to know the sources for the numbers used.

If the USGS reference is the webpage on water hardness, that reference is inappropriate because (1) it is derivative of earlier works from the 1970s (from Briggs, J.C., and Ficke, J.F., 1977, Quality of Rivers of the United States, 1975 Water Year -- Based on the National Stream Quality Accounting Network: USGS Open-File Report 78-200), so it's not a 2016 reference, and (2) the USGS has collected much more data on trace elements and calcium more recently than 1975!!

Morr (2016) is also somewhat irrelevant -- The study design was limited to samples of bottled water, which are biased toward sources that are known to be particularly clean: "Most spring waters were found to have a relatively low calcium concentration, with an average of 21.8 mg/L." Why are we comparing to the average in a biased set of spring waters that are low in calcium?

Lines 179-180: The "Approaching" column in table 3 is meaningless since we don't know which basis for comparison was used - it could be either the EPA MCL or a different concentration number from one of the 3 publications given in lines 175-177. In addition, the text explained that some wells were sampled more than once, so it's difficult to tell whether this table is saying 4 wells of 124 wells had calcium in the poorly defined "Approaching" category or 4 samples in a number of samples that is greater than 124, since some wells were sampled more than once.

Line 183: The "Exceeding" category also is meaningless, if the basis for comparison is either the regulatory limit or the one of multiple national averages. In addition, we would expect approximately half of the water samples to have concentrations greater than the average, so how it the average a meaningful benchmark? I think it is problematic to present the results with a mixture of benchmarks that are based on human-health effects and national averages that come from inappropriate sources.

Line 184: What do the authors mean by "were selected to contrast"? Contrast with what? (Do you mean that they were selected for additional discussion?)

Table 3:  Lots of comments on this table!

The use of superscript footnotes on the elements is very confusing! If the superscript corresponds to "US Average," which source was used, the ATSDR, USGS, or Morr? I would rather see a separate column heading for "Source of Benchmark," so that readers can identify which publication was the source of the numbers used as the basis for comparison. It's frustrating that the references for these 3 publications were not provided, so there is no way for this reviewer to verify the numbers used on table 3.

I dislike the use of the term "Exceeding" in this context, since the term "exceedance" has legal connotations; I would prefer to see "Concentration Greater Than Benchmark."

I get the impression that the authors were too terse in the column headings because of a perception of space constraints. What does "Below" mean? Does it mean "Concentration Less Than Benchmark"? The column heading "Regulatory Guideline" is misleading, since national averages are not regulatory. The heading "Below Detect" should be "Concentration Less Than Detection Limit." Why doesn't the category for "Below" include the concentrations that are less than the detection limit, since non-detections are likely to be less than the benchmark concentrations?

What were the detection limits? Please provide them. It's meaningless to know how many samples had concentrations "below detect" without knowing the detection limits.

Per the footnotes... What is a US EPA "Tertiary" guideline? I can't find that term used in the context of drinking-water regulations. Please define, or replace it with an accepted term.

V: I do not see a Secondary guideline for Vanadium of 0.2 µg/L. V is not listed at https://www.epa.gov/dwstandardsregulations/secondary-drinking-water-standards-guidance-nuisance-chemicals#self. There is a reference concentration of 21 µg/L used as a benchmark for comparison for the Unregulated Contaminant Monitoring Rule (UCMR): https://www.epa.gov/sites/production/files/2017-02/documents/ucmr3-data-summary-january-2017.pdf. I see a requirement for a minimum reporting level of 0.2 µg/L for vanadium for the UCMR program, but that's a laboratory capability requirement, not a guideline for human health. [Note that the ATSDR uses the term “MRL” for a different purpose (i.e., to describe “Minimal Risk Levels”). The UCMR term and ATSDR term have no relationship to each other.]

Ca: Where did the 21.8 mg/L come from? This number is unrealistically low. Can't tell without more information. I found it in Morr (2016), but only because I guessed right.

Li: I do not see a Secondary guideline for Lithium. It's not on the EPA table for Secondary drinking-water standards. I searched every reference I could think of, but could not find a source for the 73 µg/L.

Al: It is confusing to present a benchmark as a range of numbers. Does this mean concentrations in water should be within this range? Are the numbers in the other columns based on Al <50 or <200 µg/L? Which number was used as the basis for comparison? The World Health Organization recommends that Al not be greater than 200 µg/L. 

Mo: I do not see a Secondary guideline for Molybdenum. It's not on the EPA table for Secondary drinking-water standards. I was able to find a 1-day and 10-day health-advisory level (HAL) of 80 µg/L for a 10-kg child, but the more appropriate level to use would be the lifetime HAL of 40 µg/L (https://www.epa.gov/sites/production/files/2018-03/documents/dwtable2018.pdf). 

Sr: I do not see a Secondary guideline for Strontium. It's not on the EPA table for Secondary drinking-water standards. There is lifetime HAL of 4mg/L (https://www.epa.gov/sites/production/files/2018-03/documents/dwtable2018.pdf), so the benchmark should be identified as the lifetime-HAL.

Ni: Where did the national average of 3-10 µg/L come from? It seems implausible that 97.6 percent of the water samples would be less than the national average; this makes me suspect that the average came from a source this isn't relevant for this purpose. These numbers must be traceable back to their sources. It is confusing to present a benchmark as a range of numbers. I can't tell which number (<3 or <10) was used for the rest of the columns. There is a lifetime-HAL for nickel (100 µg/L); I'm not sure why that wasn't used. See https://www.epa.gov/sites/production/files/2018-03/documents/dwtable2018.pdf

Sn, Be, etc., through Pb: I see many other discrepancies on table 3 similar to the ones noted so far for the other elements (for instance, lead/Pb has an action level of 0.015 mg/L, not zero), and I feel that it's beyond the scope of my role as a peer reviewer to identify all of these issues. Given the amount of new content that would need to be provided, I suggest that Table 3 be resubmitted for peer review following revision.

Did you consider evaluating the samples for total dissolved solids, which can be approximated by converting all of the elements to the same units (mg/L) and summing the concentrations? (There is an EPA SMCL for TDS at https://www.epa.gov/dwstandardsregulations/secondary-drinking-water-standards-guidance-nuisance-chemicals#self)

Line 193: "The median of the six selected elements demonstrate a right-skew..." -- Actually, all of the constituents (except Hg and maybe Sb) show a right skew, because all of the medians are markedly lower than the respective averages. This is to be expected because most constituents in the environment do not have normal distributions. All of the comparisons will need to be re-evaluated after revising Table 3. After doing so, I suspect that V and Ca won't rank in the top 6.

Line 198. The assertion that the sites with the particularly high concentrations can attributable to evaporation warrants more discussion, since the "troughs, springs, and tanks" would all be influenced by evaporation (and probably also by dusts). Is there any statistical differences in the populations of sites where the water is exposed to the open air (vs. covered) and higher (vs. lower) concentrations?

Table 4: This table has many of the same issues as were identified for Table 3, including the problems with Benchmarks being inappropriate or not clearly traceable to literature sources. In addition, I believe some of the results are reported with inappropriate numbers of significant digits. For example, the Al result of 64,555.12 µg/L is implying a precision of ±0.01 mg/L in a result that is 1 million times as large, which is unrealistic. Table 4 lists a new type of reference (WHO Drinking Advisement) that wasn't mentioned in the text or in Table 3, which is concerning. All of the results on table 4 should be revised as needed, based on any revisions to the benchmarks selected for table 3.

Line 204. The assertion that V has the most exceedances is particularly problematic, given that the benchmark that was used for V is likely the minimum detection limit required by the EPA UCMR program, which is not an appropriate basis for comparison. It's not surprising that many samples had concentrations greater than the recommended detection limit - the detection limit is supposed to be a very low concentration.

Table 5. The rationale for which elements were selected for additional discussion might fall apart after revisiting the benchmarks on table 3. Specifically, the discussion of vanadium and calcium are problematic, given that the so-called "regulatory limits" for these elements are not regulatory limits at all. Please revise table 5 after making any appropriate revisions to table 3 and 4.

Line 211: I don't think any of the results in table 5 were interpreted relative to USGS averages. The only USGS report cited was the inappropriate reference for calcium, and the benchmark for calcium was taken from Morr (2006), which is also inappropriate. 

Lines 218-232: Seems strange to lead with arsenic information for places other than the study area. What's missing is a simple statement on the conclusions for arsenic and uranium for this study. For example, "Arsenic was detected at concentrations greater than the U.S. EPA MCL of 10 µg/L in 17 percent (40 of 235) of the sites that were sampled. Uranium was detected at concentrations greater than the U.S. EPA MCL of 30 µg/L in 9 percent (21 of 231) of sites sampled."

Lines 233-234: This statement is inaccurate. I am sure that the benchmarks used for these 4 elements were not USEPA MCLs or USGS averages.

Lines 238-239: "these elevations seem reasonable given the geologic profile of the Navajo Nation" -- what is the basis for this conclusion? Given that the benchmarks used were inappropriate in the first place, it seems like conjecture to say anything about the reasonableness of the concentrations without citing other literature about water quality in the Navajo Nation or its geology.

Lines 259-262: Please revisit this statement after addressing the comments regarding the selection of benchmarks used for this study. I can't find EPA "tertiary drinking water standards" and think such a thing doesn't exist. Please avoid coining your own terminology. Instead, refer to the EPA lifetime-HALs or UCMR reference concentrations by name.

Author Response

We appreciate the detailed review by Reviewer 1.  The following are the specific responses that were made within the manuscript.  We list the responses with respect to the original line numbers from the manuscript submitted in June to stay consistent with the reviewer’s comments.

-          17 to 18: Changed wording of “exceeding national limit” to “concentrations greater than a regulatory limit” throughout paper. Provided an additional sentence in the first paragraph of the “Results” section that clarifies although regulatory guidelines are applicable to regulated and treated water, these guidelines do serve as a basis for comparison for unregulated and untreated water consumed by humans.

-          44: Provided specific years of uranium mining activity on the Navajo Nation (1944 to 1986). Included new citation mentioned by reviewer.

-          64 to 65: Provided clarification of “down the central portion of…” with the addition of “north to south.”

-          72 to 73: Restructured the sentence to provide clarification that the recent interests in water contamination is due to research planned to assess the impact of possible future uranium mining around the Grand Canyon.

-          77: Modified to read:  The information presented in this paper will serve to provide information to Navajo Nation leaders….  We agree with the reviewer and removed the comment on regulatory agencies

-          98: Included sentence and citation explaining the circles of differing size, color, and score are based on EPA’s Hazard Ranking System, including a new citation to EPA’s explanation of their ranking system. Included specific name for samples represented in yellow as “Ingram Lab Samples.” Map legend was not changed because they are labeled as “Ingram Lab Samples” opposed to “Ingram.”

-          105: Provided clarification that the sources come from unregulated ground water and the samples were “accessed” by “windmills, troughs, etc.” Changed “wells” in Table 3 to “Total Water Sources Sampled.” Table 4 makes no mention of “wells.”

-          110-113: We clarify the convenience factor of hauling from unregulated sources on line 51 of the introduction/background, and it is mentioned low availability of regulated water to Navajo community members on lines 254-560 of the “Results and Discussion”.

-          122: Provided clarification for situations of why water resource was not run prior to collection in the presence of community members or if water source was low.

-          123: Deleted the word “total.”

-          126-127: Removed mention of sample that had an aliquot of KMnO4 in the field. The purpose of this was to compare the suggested EPA method that did not add KMnO4 with an adjusted method that did to determine if this impacted the concentration of Hg in the sample. No statistically significant change was observed in measurement, both samples had background levels of Hg in the sample and a spiked field sample had statistically similar concentrations between each sample. For these reasons and to avoid transportation of additional chemicals in the field, it was elected not to add KMnO4 in the field. The last paragraph in Methods – 2.3 Sample Analysis and Instrumentation has been adjusted to reflect this change.

-          158-161: See explanation to previous comment.

-          165-166: Provided clarification that QCs were re-analyzed throughout all analyses.

-          167: Added “split” to the 121 subsamples sent off to explain the nature/type of samples that were sent off to certified laboratories.

-          Line 168: Removed the adjective “certified” from description of the service laboratories used. Certification language typically refers to a single method opposed to a blanket statement for the laboratory.

-          Line 175-177: Referring to Table 1, the three elements in question are Ca, Ni, and Sn. These elements do not have either a primary or secondary US EPA maximum contaminant level (MCLs), so national (USA) averages were used from alternative sources which were/have been cited.

Calcium: The reviewer pointed out correctly the calcium number was derived from a 1975 USGS report as well as a research paper by Morr (2006). The USGS acknowledges that this information from their 1975 is dated, but comments “Although this map illustrates data from 1975, these data have been found to be accurate and useful in current assessments” updated 30 July 2018 (https://www.usgs.gov/special-topic/water-science-school/science/hardness-water?qt-science_center_objects=0#qt-science_center_objects). As for the Morr (2006) paper, the authors state “…Water calcium concentrations were collected and classified into purified waters (from municipal aqueducts), spring waters (originating in a free-flowing spring)…”. The USGS states that spring water comes from aquifers underground, implying groundwater as the source. (https://www.usgs.gov/special-topic/water-science-school/science/springs-and-water-cycle?qt-science_center_objects=0#qt-science_center_objects). Given these two pieces of information, the reported “national average” used for the paper was not changed.

Nickel: On the provided ATSDR reference (https://www.atsdr.cdc.gov/ToxProfiles/tp15-c2.pdf), the third paragraph of section “2.1 Background and environmental exposures to nickel in the United States” states “Nickel concentrations in surface water and groundwater range between 3 and 10 ug/L.” ATSDR has a narrower range for treated drinking water in the United States, however because the water supplies for this study are all surface and groundwater (which has been amended in the paper to reflect), the authors felt using the average for surface and groundwater was more representative. For this reason, the nickel concentration used for the paper was not changed nor was the reference updated.

Sn: A reference from the World Health Organization was added that provides the statement “A mean range of 1.1-2.2 ug/litre (maximum 30 ug/litre) was found in a survey of water supplies in the USA.” The reported average in Table 1 has not changed, but a reference has been added demonstrating where the number originated from.

None of the language within the paper has been changed, because like US EPA primary and secondary MCLs, the references are provided for the reader to search out and understand why these concentrations are set: it is not the purpose or intent of the paper to justify or critique guidelines, national averages, or how these numbers were determined.

-          Line: 179-180: No change was made to “approaching” on column 3. The first paragraph of the results section explains where guideline numbers were used to base comparison off: EPA primary and secondary MCLs where available and USA national averages reported in other sources (with the sources provided, see response to comment above). Regarding the reviewer’s comment that some samples were sampled more than once, the paper explains that for any sample that was sampled and analyzed more than once, only the final average was reported in this paper and are treated as a “sample.”

-          Line 183: This comment was addressed from a previous comment: the language was changed from “exceeding” to “concentrations greater than a guideline…”. Regarding these guidelines being arbitrary, as the reviewer’s previous comment (and the changed language in the paper reflects) suggested, these guidelines are only for regulated and treated water, of which none of these samples are; however, national averages and regulatory guidelines do serve as a useful basis to make comparisons from in the absence of another guideline for unregulated water supplies. The language explaining why six elements were selected for closer evaluation was not changed. The selection of these elements when many exceeded a national primary regulatory guideline seems appropriate and is in line with the reviewer’s previous comment that the guidelines serve as a basis for comparison. For the elements without a primary MCL or that used a national average, the ATSDR resource listed gave information that despite a lack of primary MCLs, some elements have been associated with detrimental human health effects. Given these samples have numerous exceeding the guidelines, the authors of this paper felt it appropriate to provide this information, especially as this information has been and will be provided to the communities impacted.

-          Line 184: Changed language from “contrast” to “expand upon.”

-          Table 3:

o   Added column “Source of Guideline” to provide information where guideline was acquired and removed superscripts.

o   Added “Guideline Sources” in figure caption to provide detail of abbreviations used in table. Removed superscripts on elements that previously provided this information.

o   Specified that the lower limit was used for comparison on the three elements that had a range in the literature.

o   All guidelines used for the table were verified by the sources listed and new references have been added as necessary (US EPA NSDWR, US EPA UCMR, US EPA RSL, US EPA DWSHA).

o   Changed the guideline for Vanadium from 0.2 ug/L to 21.0 ug/L. Adjusted numbers of samples in respective categories on table to reflect this change. These changes were made to Table 5 and within the body of the text to reflect this change as well.

o   Added detection limits to supplemental information.

o   Updated lithium’s guideline to US EPA Regional Screening Levels. Updated Tables 3 and 4 to reflect this change.

o   No change was made to “Below” versus “Below Detect.” The paper makes this distinction in the methods, however, a reading of “Below” is still able to be accurately and reliably read by the method and instrument used. In comparison, a value below an instrument or method’s detection is marked to determine that if there was a concentration in the sample, it cannot be commented on analytically.

o   Total dissolved solids measures were considered; however, because the protocols used in the EPA method called for the filtering of all samples it was decided reporting as absolute concentration was more representative. Comparing filtered/non-filtered and different water preparation methods (digestions for whole, unfiltered water that could report total dissolved solid values) was something conducted with collaborators at the University of New Mexico. It was found that there was no appreciable difference in all waters, unless the water was visibly turbid and stagnant. This type of water, both groups and community partners identified are not commonly used for human consumption. For these reasons, whole concentration approaches were adopted. 

-          Line 193: Deleted statistical comment about medians.

-          Line 198: Provided clarification on variables that could have influenced contaminant concentration by, removing previous sentence about “Evaluation of the location and identification…” and replacing with sentence explaining these other variables warrant investigation.  

-          Table 4:

o   Benchmark superscripts were removed from the elements.

o   Significant figures were adjusted.

o   Citations and benchmarks updated from Table 3.

-          Line 204: Adjusting the guideline for vanadium produced 96 samples that were above the guideline used, compared to 97. The number has changed, but many samples still were above this guideline.

-          Table 5: Vanadium values were re-evaluated with the revised guideline and many samples still above this guideline. The guideline to which calcium was compared to was defended in a previous comment. Due to this, calcium’s inclusion for further discussion was maintained. Changed wording in figure caption from “regulatory limits” to “guideline comparisons.”

-          Line 211: See previous comment(s) on calcium’s guideline and inclusion in Table 5.

-          Lines 218-232: Added introductory statement about arsenic’s occurrence for this study, before commenting on arsenic’s occurrence across the United States and the Southwest.

-          Lines 233-234: Changed “US EPA MCLs” to “US EPA Guidelines.” See previous comments regarding USGS calcium average.

-          Lines 238-239: Provided citation for calcium and vanadium in drinking water and geology of the Navajo Nation.

-          Lines 259-262: Changed language from “secondary and tertiary US EPA…” to “non-enforceable guidelines and national averages.” Change “limits” to “guidelines.” 

Reviewer 2 Report

An interesting work adding useful information for the authorities and society. Some corrections/additions are recommended in order the paper to be accepted (see attachment)

Author Response

We appreciate the detailed review by Reviewer 2.  The following are the specific responses that were made within the manuscript.

-          Agreed with reviewer. At the start of the Methods Section in “2.1 Study Area,” inserted geological description of the Navajo Nation with focus on hydrology.

-          Site locations and descriptions added in Supplementary Materials Section.  Please note that this information is detailed so the spreadsheet provided is fairly large.  It may be too much detail, but we have included it for the editors to decide to include or exclude it as supplementary material.

-          Citation numbers added to the end of all instances where a researcher’s paper demonstrated a finding: EX: In one study by Hoover…

Reviewer 3 Report

The review report of the manuscript: 'Quantification of Elemental Contaminants in 2 Unregulated Water across Western Navajo Nation'. The paper includes valuable results. Topic of the paper is interesting however, concerning a local scale. A subject of investigations  is suitable for journal of International Journal of Environmental Research and Public Health although not novel. The research on  determination of arsenic and uranium in undergrand water for quantificationof other possible elemental contamination that may exist  have been carried out from 90 years of the last century. 

This paper is interesting but there is no section discussion of the results and the only report. This paper in my opinion, has no signs of trying to solve the problem or explain the obtained results. The section conclusions is too long and there are not conclussions but a kind of discussions. Only the last paragraf can be considered as a conclusion from the conducted research.

General comments about findings that need revision

- sections 3 Results and Discussion; should be corrected throughout ;

- section 4 Conclusions are not reported enough, therefore the reader cannot evaluate the data.

Author Response

We appreciate the detailed review by Reviewer 3.  The following are the specific responses that were made within the manuscript.  

We agree with reviewer 3 regarding a need for stronger and concise conclusion. This was added to the end of the paper. Discussion section merged with Results section.

Round 2

Reviewer 1 Report

The revised paper is much improved; however, I am still not sure that the Vanadium data are presented correctly. The reference concentration is given as 21 micrograms per liter in table 1, and table 1 says 97 of the 121 samples had concentrations "above" this concentration. However, table 2 says that the median concentration was 2.73 micrograms per liter. Maybe there are differences in the datasets, but it seems odd that the median is 2.73 compared to 21, but 97 of 121 are supposedly above 21? The median is the 50th percentile, so how can the median be 2.73, if 97 of 121 samples have concentrations greater than 21 micrograms per liter?

I'm also concerned about referring to concentrations that are above average as "exceedances," as in line 245. Being above the average concentration isn't really an exceedance, in my opinion. Half of any population is greater than the median. 

The number of vanadium samples is 119 in line 245, but listed as 121 in table 1. 

Please double-check all of the numbers for consistency and accuracy. Thanks.

If there are discrepancies for good reasons, please list the reasons.

Author Response

-          The reviewer is correct. The Excel Spreadsheet that was used to calculate the median and average was incorrect (incorrect cells selected), resulting in a lower median and average. This has been changed to reflect the correct number.

-          Changed the wording from “exceedances” to “above the comparative guideline” within the body of the text.

-          The reviewer is correct, it was overlooked on last revision to change the numbers and reflect them accurately. Per the reviewer’s advice, all numbers have been verified and discrepancies screened for. No additional errors were detected, beyond the issues with the Vanadium numbers.

o   The complete master database containing all of the data, in addition to other data, will be included in a future publication that contains more than 8000 water locations and more than 20,000 data points.

Reviewer 3 Report

I appreciate the effort put in improving the manuscript.

In my opinion in its present form it may be published in the IJERPH.

Author Response

We appreciate Reviewer 3's suggestions for making the manuscript stronger.  We also appreciate Reviewer 3 re-reading the revised manuscript.